# Bioactive Compounds and Antioxidant Activity of Mango Peel Liqueurs (*Mangifera indica* L.) Produced by Different Methods of Maceration

**DOI:** 10.3390/antiox8040102

**Published:** 2019-04-16

**Authors:** Emanuela Monteiro Coelho, Marcelo Eduardo Alves Olinda de Souza, Luiz Claudio Corrêa, Arão Cardoso Viana, Luciana Cavalcanti de Azevêdo, Marcos dos Santos Lima

**Affiliations:** 1Department of Food Technology, Instituto Federal do Sertão Pernambucano, Petrolina PE 56314-522, Brazil; emanuela-monteiro@hotmail.com (E.M.C.); marcelo.olinda@ifsertao-pe.edu.br (M.E.A.O.d.S.); arao.viana@hotmail.com (A.C.V.); luciana.cavalcanti@ifsertao-pe.edu.br (L.C.d.A.); 2Empresa Brasileira de Pesquisa Agropecuária, Semiárido, Petrolina PE 56302-970, Brazil; luiz.correa@embrapa.br

**Keywords:** phenolic compounds, pectinase, HPLC

## Abstract

The present work had the objective of producing liqueurs from mango peels (varieties “Haden” and “Tommy Atkins”) by processes of alcoholic maceration and maceration with pectinase, as well as to evaluate bioactive compounds by reversed-phase high-performance liquid chromatography coupled to diode array detection and fluorescence-detection (RP-HPLC/DAD/FD) and in vitro antioxidant activity (AOX), for by-product potential reuse. Alcoholic maceration in wine ethanol (65% *v*/*v*) produced liqueurs with higher phytochemical and AOX content. Maceration with pectinase resulted in liqueurs with higher quercetin-3-*O*-glucopyranoside content. In relation to mango varieties, Haden liqueurs presented higher bioactive content than Tommy Atkins liqueurs. The liqueurs presented high antioxidant activity. The main bioactive compounds found were flavanols (epicatechin-gallate, epigallocatechin-gallate), flavonols (quercetin-3-*O*-glucopyranoside and rutin), and phenolic acids (gallic acid, *o*-coumaric acid, and syringic acid). The present study showed that the production of liqueur enabled the recovering of an important part of the bioactive content of mango peels, suggesting an alternative for the recovery of antioxidant substances from this by-product.

## 1. Introduction

Mango (*Mangifera indica* L.) is one of the most popular edible fruit and its production, at present, ranks seventh in world fruit production, just after watermelons, bananas, apples, grapes, oranges, and coconuts [1]. Mango varieties, such as “Haden” and “Tommy Atkins”, are appreciated for having attractive color, characteristic taste, and various nutrients beneficial to health [2]. Mangoes are a potential source of flavonoids and carotenoids, which makes them a food with good bioactive potential [3,4,5]. The main bioactive compounds found in mangoes are polyphenols mangiferin, catechins, quercetin, kaempferol, gallic acid, and benzoic acid, which are compounds associated with the prevention of degenerative diseases, including cancer, cardiovascular diseases, and diabetes [4,6]. 

Mangoes are generally consumed in their natural form but can also be processed to obtain pulps and juices, a practice that generates large amounts of waste [7]. The peels are an important by-product, little used for processed foods, and which have potential of being a functional ingredient [1]. According to Liu et al. [5], mango peels contain a considerable concentration of bioactive compounds, and these authors suggest that the rational use of these residues can bring, in addition to nutritional benefits, a reduction in environmental impacts. 

One way of obtaining processed foods from fruit peels is by producing liqueurs [8]. The maceration of the fruit in ethanol or distilled beverages favors the extraction of bioactive and aromatic compounds present in the raw material [8,9,10]. Traditionally liqueurs are alcoholic beverages obtained by maceration of pulp or fruit peels in ethanol solutions for periods ranging from 15 to 30 days. Subsequently, a sugar syrup is added to the extract obtained, which is then filtered and stored to mature for periods that normally range from 60 to 240 days, followed by further filtration and bottling [10,11]. 

One of the techniques to optimize the extraction of compounds present in fruit peels is the addition of pectinases enzymes in the maceration process [12]. The addition of pectinases in fruit maceration degrades the pectin present in the peels and facilitates the release of compounds, resulting in a better extraction of the bioactive compounds [13]. This also causes modifications in the profile of phenolic compounds present in the fruit [14]. 

Mango peels are one of the main by-products of the mango industry, and studies have shown that this material contains phytochemical compounds of nutritional interest. One of the ways to take advantage of fruit peels to obtain processed products is in the manufacture of liqueurs. In this context, the present work had the objective of producing liqueurs from mango peels (varieties “Haden” and “Tommy Atkins”) by processes of alcoholic maceration and maceration with pectinase, as well as to evaluate bioactive compounds by HPLC and in vitro antioxidant activity, in order to reuse this by-product.

## 2. Materials and Methods 

### 2.1. Standards and Reagents

Wine alcohol 65% *v*/*v* was obtained from winery Ouro Verde-Miolo Wine Group (Casa Nova, BA, Brazil). Folin-Ciocalteu reagent, ethanol, potassium persulfate, phosphoric acid, and potassium phosphate monobasic were purchased from Merck (Darmstadt, Germany). Trolox (6-hydroxy-2,5,7,8-tetramethylchromate-2-carboxylic acid) and the 2,2-diphenyl-1-picrylhydrazyl (DPPH) radicals and 2,2-azino-bis (3-ethylbenzthiazoline-6 sulfonic acid) (ABTS) were supplied by Sigma-Aldrich (St. Louis, MO, USA). Methanol was from J.T. Baker (Phillipsburg, NJ, USA). Ultrapure water was obtained with a Marte Científica purification system (São Paulo, SP, Brazil). External standards of gallic acid, syringic acid, benzoic acid, *p*-coumaric acid, *o*-coumaric acid, cinnamic acid, procyanidin B_1_, catechin, procyanidin B_2_, and caffeic acid came from Sigma-Aldrich (St. Louis, MO, USA). Epigallocatechin gallate, epicatechin, epicatechin gallate, procyanidin A_2_, quercetin 3-glucoside, rutin, kaempferol 3-glucoside, myricetin, and isorhamnetin 3-glucoside, quercetin-3-*O*-glucopyranoside came from Extrasynthese (Genay, France). The *trans*-Resveratrol was obtained from Cayman Chemical Company (Michigan, EUA, Ann Arbor, MI, USA).

### 2.2. Raw Material

To elaborate the liqueurs, “Haden” and “Tommy Atkins” mangoes were used, originating from the São Francisco Valley in the Northeast of Brazil. The fruits were harvested in March 2016, at full ripeness stage. The meteorological conditions at harvest were: rainfall 0.0–4.2 mm; average temperature 29.9 °C; relative humidity 51%; solar radiation 462.2 ly/day; and insolation 8.5 h/day. Meteorological measurements were taken at the agro-meteorological station of Bebedouro (Petrolina, Pernambuco State, Brazil, 09°09′ S 40°22′ W).

### 2.3. Production of the Liqueurs

The mangoes were sanitized in chlorinated solution (50 ppm), then peeled manually. The peels were cut into 1.0 × 1.0 cm and used to obtain liqueurs by alcoholic maceration and maceration with pectinase.

*Alcoholic maceration*: Mango peels (1000 g) were placed to macerate in one liter of wine alcohol (65% *v*/*v*) where they remained for 30 days at room temperature. After maceration the liquid was drained (extract) and mixed with sucrose syrup until a standardized liqueur containing 15% total sugars and 18% ethanol was obtained. 

*Maceration with pectinase*: The peels (1000 g) were mixed in one liter of water with 100 μL of pectinase (Endozym^®^ Pectofruit PR made by Spindal-Pascal Biotech (Gretz-Armainvilliers, France)). The maceration was carried out at 60 °C for 90 minutes. After maceration, the extract was drained and cooled to room temperature. To obtain the liqueur, a mixture of sucrose syrup and wine alcohol (65% *v*/*v*) was added to the extract to provide a beverage containing 15% total sugars and 18% alcohol.

For the standardization of the sugar content and alcoholic degree of the liquors, mass balance calculations were performed to measure the sugar and alcohol contents in the obtained extracts, and at the final concentrations desired. The liqueurs were packed in 300 mL colorless glass bottles, manufactured by Saint-Gobain^®^ (São Paulo, SP, Brazil), and placed to mature for 180 days in an environment protected from light and at a temperature of 25 ± 3 °C. The treatments consisted of producing four liqueurs from mango peels, denominated as: Haden by alcoholic maceration (HDA), Haden by maceration with pectinase (HDP), Tommy Atkins by alcoholic maceration (TAA), and Tommy Atkins by maceration with pectinase (TAP). Each treatment was performed in three replicates, totaling 12 produced liqueurs.

### 2.4. Physicochemical Analysis and Color Measurement by CIEL*a*b* System

To obtain the basic analytical characteristics of the liqueurs, classical pH analyzes were carried out; alcohol (%*v*/*v*), titratable acidity, and total sugars %, following the methodologies described in Association of Official Analytical Chemists International (AOAC) [15]. Color analysis was also conducted using a MiniScan EZ digital colorimeter (Hunterlab, Reston, VA, USA) using L*, a*, and b* systems, where L* denotes lightness and ranges from 0 (black) to 100 (white), and a* and b* denote opponent dimensions, ranging from green (−) to red (+) and from blue (−) to yellow (+), respectively.

### 2.5. Determination of the Bioactive Compounds Profile by RP-HPLC/DAD/FD

The individual phenolic compounds in the liqueurs were determined by reversed-phase high performance liquid chromatography (RP-HPLC) on a Waters Systems instrument (model Alliance e2695) coupled to diode array detection (DAD) and fluorescence detection (FD). For the separation of compounds, a Gemini NX C-18 column (150 mm × 4.6 mm × 3 μm) and a Gemini NX C-18 guard column (4.0 mm × 3 mm × 3 μm) were used, both manufactured by Phenomenex (Torrance, CA, USA). The gradient used was 0 min: 100% A; 18 min: 87.5% A, 2.5% B, 10.0% C; 30 min: 83.5% A, 3.2% B, 13.3% C; 36 min: 75.0% A, 5.0% B, 20.0% C; 48.5 minutes: 65.0% A, 8.3% B, 26.7% C; 50 min: 65.0% A, 8.3% B, 26.7% C; and 65 min: 100% A. Solvent A consisted of a solution of 25 mmol L^−1^ of potassium dihydrogen phosphate with the pH adjusted to 2.05 with phosphoric acid, solvent B was methanol, and solvent C was acetonitrile. The oven temperature was maintained at 40 °C and the solvent flow at 0.6 mL min^−1^, with a total run time of 65 min. The detection and quantification of the compounds was carried out using external standards. 

The analysis was performed according to the methodology described by Natividade et al. [16], using the software program Empower™ 2 (Milford, MA, USA) for data treatment. 

### 2.6. Total Bioactive Content and In Vitro Antioxidant Activity

The total phenolic content of the liqueurs was determined by the spectrophotometric method with Folin-Ciocalteu [17]. The absorbances of the samples, read at λ = 765 nm, were compared with a calibration curve obtained with gallic acid, with the results being expressed as μg equivalent to gallic acid per 100 mL of liquor (µg 100 mL^−1^ GAE). 

To measure the antioxidant activity in vitro, we used the methods of free radical scavenging with DPPH (1,1-diphenyl-2-picrylhydrazyl) and ABTS 2,2-azino-bis (3-ethylbenzthiazoline-6-sulfonic acid) [18,19]. The Trolox analytical standard was used to build the calibration curves and the results were expressed as Trolox equivalents per 100 mL of liqueur (µmol TEAC 100 mL^−1^). For the analysis, solutions of DPPH 1.0 mmol and ABTS radicals were prepared in ethanol and diluted to an absorbance of 0.90 ± 0.05 and 0.70 ± 0.03, respectively. The antioxidant activity of samples was assessed through the rate of decay in absorbance at 517 nm for DPPH and at 754 nm for ABTS. In the DPPH method, absorbance was measured at time *t* = 30 min after the addition of liquors. In the ABTS method, absorbance was determined at time *t* = 6 min after the addition of samples. All the analyses were performed in triplicate. Absorbance readings were performed in a spectrophotometer UV-Vis 2000A Instrutherm^®^ (São Paulo, SP, Brazil).

### 2.7. Statistical Analysis

The results obtained from the analysis of the samples were submitted to analysis of variance (one-way ANOVA) and compared using the Tukey test at 5% of error probability using a SPSS program, Version 17.0, statistical package for Windows (SPSS, Chicago, IL, USA). A principal component analysis between the phytochemical profile and antioxidant activity was realized. 

## 3. Results and Discussion

### 3.1. Quality Parameters

The results of the classical analyses of the liqueurs are presented in Table 1. The maceration methods studied produced mango liqueurs with different pH values and titratable acidity. The beverages obtained by maceration with pectinase had lower pH values (3.61–3.94) and higher titratable acidity (4.4–4.6 g L^−1^), while maceration in ethanol resulted in liquors with pH and acidity of 4.84–5.01 g L^−1^ and 1.0–1.1 g L^−1^, respectively. The lower values of pH and higher values of titratable acidity in the liqueurs obtained by maceration with pectinase can be explained by the higher solubility of organic acids of the peels in water (solvent of maceration with pectinase), as citric acid and malic acid [20]. A higher titratable acidity in liqueurs can favor a better taste balance, represented by the ratio of sugar/acidity [11]. 

There were no significant differences in the values of total sugars and alcoholic strength of the liqueurs produced, with this result being expected due to the standardization performed in the elaboration of beverages. Regarding color, *L** values varied between 55–58.7 and 62.3–70; *a** varied between 8.2–8.6 and 6.1–1.2; and *b** varied between 80.7–63.3 and 71.5–37.6, for liqueurs obtained by maceration in ethanol and pectinase, respectively. The liqueurs obtained by maceration in pectinase had lower values of *L** and *a**, and higher values of *b**, which represents a darker yellow color than the liqueurs obtained with ethanol.

### 3.2. Total Phenolic Content and Bioactive Profile in Mango Peel Liqueurs

The total phenolic content and bioactive profile pertaining to the families of flavanols, flavonols, stilbene, and phenolic acids in the liqueurs are shown in Table 2. The liqueurs obtained by maceration in wine alcohol (65%) obtained a higher total phenolic content in relation to those obtained with pectinase, with values varying from 64,787 to 70,564 μg 100 mL^−1^ for the Tommy Atkins and Haden varieties, respectively. A comparison of the preparation methods showed that there was no significant difference between pectinase and ethanol for the Haden variety; for the Tommy Atkins variety, however, the maceration method in ethanol obtained a higher total phenolic content. In relation to the total phenolics quantified by HPLC, the liqueurs obtained in wine alcohol also presented higher values, especially the Haden variety, with a total of 10,523 μg 100 mL^−1^.

The total phenolic content obtained in the Tommy Atkins and Haden mango peel liqueurs was considered high in relation to the study by Sellamuthu et al. [4], who reported values ranging from approximately 30,000 to 76,430 μg 100 g^−1^ in mango varieties “Peach”, “Phiva”, “Rosa”, “Saber”, “Tommy Atkins”, and “Zill”. In the work of Sultana et al. [21], the total phenolic content in “Langra” and “Chonsa” mango peels ranged from 116.8 to 122.6 mg g^−1^ dry mass. The values obtained for total phenolics by the Folin-Ciocalteu method and phenolic total quantified by HPLC showed that the production of this beverage from mango peels considerably recovered the bioactive content of this fruit. 

#### 3.2.1. Flavanols

The mango peel liqueurs obtained by alcoholic maceration, in general, presented higher values for sums of quantified flavanols by HPLC, where the values varied from 950 to 1254 μg 100 mL^−1^ for the Haden and Tommy Atkins varieties, respectively (Table 2). In liqueurs obtained by maceration with pectinase, the sum of flavanols ranged from 541 to 581 μg 100 mL^−1^ for Haden and Tommy Atkins, respectively. In all liqueurs, epicatechin gallate was the major flavanol quantified, with mean values ranging from 266.2 to 718.7 μg 100 mL^−1^. The second main flavanol present in the analyzed liqueurs was epigallocatechin gallate, with values ranging from 64 to 174 μg 100 mL^−1^. In smaller values (μg 100 mL^−1^), the following were also quantified: catechin (8–141), epicatechin (13.3–22), procyanidin A_2_ (14–78), procyanidin B_1_ (29.3–88), and procyanidin B_2_ (0–10).

These values obtained for flavanols in the liqueurs were in line with those reported in the work of López-Cobo et al. [1], who found Catechin values of 11.1, 6.4, and 27.9 mg 100 g^−1^ in dry extract of “Keitt”, “Osteen”, and “Sensación” mango varieties, respectively. In the study of Dorta et al. [22], different solvents were used to extract flavonoids from mango peels, noting that the best solvent for obtaining antioxidant flavonoids in this by-product was ethanol 50%, which may explain the higher values obtained for flavanols in liqueurs produced by maceration in wine alcohol in the present study. 

#### 3.2.2. Flavonols and *Trans*-Resveratrol

In relation to flavonols, liqueurs obtained by maceration with pectinase presented a higher value for sum quantified by HPLC (Table 2). Quercetin-3-*O*-glucopyranoside was the predominant flavanol in the liqueurs obtained with pectinase, in values ranging from 343.3 to 348.7 μg 100 mL^−1^, and was also the substance responsible for the highest total of flavonols on the liquors obtained with pectinase. For *trans*-resveratrol, the values obtained were low, varying from 2.0 to 4.0 μg 100 mL^−1^, and practically did not differ between the liqueurs produced. For liqueurs obtained with wine alcohol, the quantified flavonols were rutin (39.3–58 μg 100 mL^−1^), kaempferol 3-glucoside (30.7–46 μg 100 mL^−1^), isorhamnetin 3-glucoside (24–30.7 μg 100 mL^−1^), quercetin-3-*O*-glucopyranoside (14–42.7 100 mL^−1^), myricetin (8–13 μg 100 mL^−1^), and quercetin 3-glucoside (6.0 μg 100 mL^−1^). Comparing the mango varieties studied, in general, the differences in the profiles of the flavonols were not relevant.

Several flavonols were quantified by Meneses et al. [23] in ethanolic extracts of industrial wastes from the processing of Tommy Atkins and Haden mangoes, and the main compounds identified, in approximate values, were: quercetin 3-galactoside (11,000 µg kg^−1^), quercetin 3-xyloside (7000 µg kg^−1^), quercetin 3-arabinoside (1800 µg kg^−1^), quercetin 3-glucoside (500 µg kg^−1^), and kaempferol 3-glucoside (100 µg kg^−1^). Based on the values of quercetin 3-glucoside and kaempferol mentioned by Meneses et al. [23] and on the values obtained for these compounds in the liqueurs of the present study, these all indicate that the production of liqueurs is a suitable technological process for the recovery of flavonols from mango peels. We also emphasize that maceration with pectinase was responsible for a considerable increase in the quercetin-3-*O*-glucopyranoside content of the liqueurs studied.

#### 3.2.3. Phenolic Acids 

Among the phytochemicals quantified by HPLC, phenolic acids were the main compounds present in the liqueurs studied, mainly in the liqueurs those obtained by alcoholic maceration, with values of 2530 and 9269 μg 100 mL^−1^ for Tommy Atkins and Haden, respectively (Table 2). The main phenolic acid present in the liqueurs was gallic acid, presenting in greater quantities in the Haden variety. Haden’s liquor was also notable for the presence of benzoic acid (1777 μg 100 mL^−1^) in the liqueur obtained by alcoholic maceration. Gallic acid has also been reported as one of the main phenolic compounds present in “Ataulfo” mango extracts, in values ranging from 43 to 62 μg mL^−1^ at different maturation stages [24]. In the study of López-Cobo et al. [1], values for gallic acid were 12.5, 13, and 26 mg 100g^−1^ (dry extract) for mango peels “Keitt”, “Osteen”, and “Sensación” varieties, respectively.

### 3.3. In Vitro Antioxidant Activity of Mango Peel Liqueurs

The antioxidant activity in vitro (AOX) of the liqueurs were measured by free radical sequestration methods with DPPH and ABTS, and the results expressed as equivalent to micromoles of Trolox per 100 mL^−1^ of liqueur (µmol TEAC 100 mL^−1^) (Figure 1). Comparing mango varieties, the Haden-produced liqueurs presented higher AOX in values of 563 and 606 μmol TEAC 100 mL^−1^ (DPPH), and 631 and 648 μmol TEAC 100 mL^−1^ (ABTS). The liqueurs obtained with Tommy Atkins presented AOX values of 256 and 447 μmol TEAC 100 mL^−1^ (DPPH), and 275 and 466 μmol TEAC 100 mL^−1^ (ABTS). Regarding the elaboration methods, the liqueurs obtained by maceration in ethanol presented higher AOX than those obtained by maceration with pectinase, for the two mango varieties studied.

According to Meneses et al. [23], the AOX (DPPH) of mango peel extracts ranged from 852 to 884 μmol TEAC g^−1^ dry mass. Comparing the values mentioned by these authors with the values obtained in the liqueurs in the present study, it is evident that the use of the peels to produce liqueurs is an alternative for the recovery of antioxidant substances from this by-product. 

The AOX values obtained in mango liqueurs (256 to 648 μmol TEAC L^−1^) were considered high, compared with a study by Leeuw et al. [25] of AOX (DPPH) of 38 commercial red wines, which are alcoholic beverages with recognized antioxidant activity. These originated from traditional regions, such as France, Italy, Chile, Argentina, and the United States, and ranged from 317 to 767 μmol TEAC 100 mL^−1^. The antioxidant activity (DPPH) of liqueurs of red fruits, such as raspberry, strawberry, cornelian cherry, blackcurrants, blackberry, sour cherry, mahonia, sloe, chokeberry, and black rose, ranged from approximately 250 to 3250 μmol TEAC 100 mL^−1^ [9], indicating that liqueurs have high antioxidant potential.

Liqueurs obtained by maceration with pectinase, even with AOX lower than with maceration in ethanol, also presented efficient antioxidant activity.

### 3.4. Principal Components Analysis

One of the ways to evaluate the influence of factors, such as processing methods in the bioactive content and antioxidant activity of the beverages, has been through chemometrics techniques, such as principal components analysis (PCA) [26]. In the present study, PCA was applied to the mango peel liqueurs as a function of phytochemical profile and antioxidant activity. Components 1 and 2 (PC1 and PC2, respectively) explained 75.9% of the variance of the experiment; PC1 explained 44.4% and PC2 31.5% (Figure 2). 

The factor analysis showed that the variables with higher contribution for the separation of the maceration treatments on PC1 with loading > 0.70 were: antioxidant activity (DPPH), antioxidant activity (ABTS), epigallocatechin (EGC), *p*-Coumaric acid (pCa), kaempferol 3-glucoside (Kae), isorhamnetin 3-glucoside (Iso), myricetin (Myr), total phenolic by HPLC (TFQ), total phenolic (TP) and procyanidin B_1_ (PB1) (positive loadings), and *trans*-resveratrol (Res) and epicatechin (Epc) (negative loadings). For PC2, the variables that contributed most to the separation (loading > 0.70) were: quercetin 3-glucoside (Que), *o*-coumaric acid (oCa), *p*-coumaric acid (Cac) and quercetin-3-*O*-glucopyranoside (QuP) (positive loadings), and rutin (Rut), epicatechin gallate (EpG), gallic acid (Gac), and procyanidin B_2_ (PB2) (negative loadings). PC1 and PC2 separated on the positive part the Haden mango peel liqueur obtained by alcohol maceration (HDA), correlated with the highest content of EGC, pCa, Cac, Kae, Iso, Myr, TFQ, TP, PB1, Que, oCa, QuP, DPPH, and ABTS, which was the liqueur that presented the highest bioactive content related to phytochemicals and antioxidant activity. PC2 separated on the negative part of the liqueurs of Tommy Atkins mango peel liqueurs (TAM) and Haden by maceration with pectinase (HDP) correlated with the higher content of Rut, EpG, Gac, and PB2. PC1 separated in the negative part of the liqueur of Tommy Atkins mango peel obtained by maceration with pectinase (TAP) associated with the higher content of Epc and Res, and lower values for the content of bioactive compounds and antioxidant activity. In general, liqueurs obtained by alcohol maceration obtained higher values of phenolic compounds and antioxidant activity in vitro, and Haden mango variety presented higher bioactive content compared to Tommy Atkins. The method of maceration with pectinases was inferior to maceration with wine alcohol 65% *v*/*v*, but this is a promising technique, and in the Haden variety it presented a satisfactory bioactive content and AOX. In a study by Cosmulescu et al. [27], different concentrations of ethanol (40–70% *v*/*v*) were evaluated in obtaining extracts to produce walnut liqueurs, reporting that ethanol 70% promoted a greater extraction of phenolic compounds, such as catechin, epicatechin, quercetin, myricetin, caffeic acid, and syringic acid, which corroborates with the results obtained in the present study.

## 4. Conclusions

The present study showed that with the use of mango peels to produce liqueurs it is possible to recover much of the bioactive content of the mango peels, which are one of the main by-products of the fruit industry. Alcoholic maceration (wine alcohol 65% *v*/*v*) produced liqueurs with higher content of phytochemicals and antioxidant activity than maceration with pectinase. Maceration with pectinase resulted in liqueurs with high values of quercetin-3-*O*-glucopyranoside. In relation to the mango varieties studied, Haden presented higher phytochemical content than Tommy Atkins. The liqueurs presented high antioxidant activity in vitro, where the main bioactive compounds found in terms of quantity were flavanols (epicatechin gallate, epigallocatechin gallate), flavonols (quercetin-3-*O*-glucopyranoside and rutin), and phenolic acids (gallic acid, *o*-coumaric acid, and syringic acid). The values obtained for phenolic compounds quantified by HPLC and antioxidant activity showed that the production of liqueur enabled the recovering of an important part of the bioactive content of mango peels, suggesting an alternative for the recovery of antioxidant substances from this by-product. 

## Figures and Tables

**Figure 1 antioxidants-08-00102-f001:**
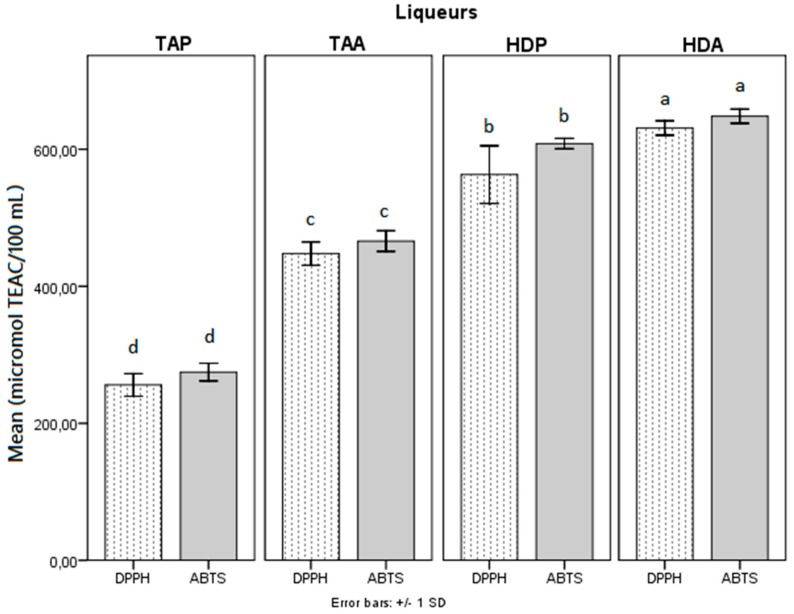
Antioxidant activity in vitro of mango peel liquors obtained by different methods. Legend: had = Haden mango peel liqueur by alcoholic maceration, HDP = Haden mango peel liqueur by maceration with pectinase, TAA = Tommy Atkins mango peel liqueur by alcoholic maceration, TAP = Tommy Atkins mango peel liqueur by maceration with pectinase. Averaged bars followed by equal letters do not differ from each other by the Tukey test at 5% error probability.

**Figure 2 antioxidants-08-00102-f002:**
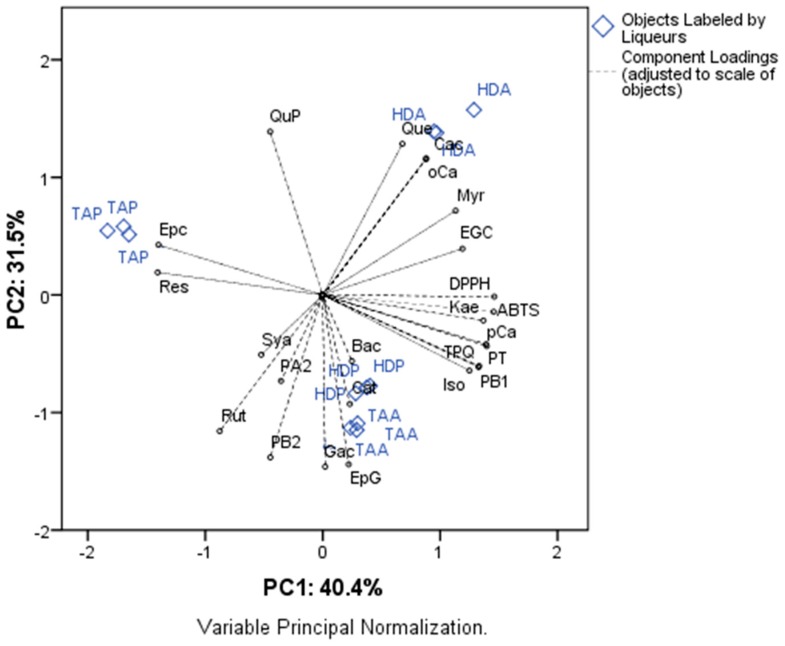
Principal component analysis between the phytochemical profile and antioxidant activity of mango liqueurs obtained by different methods. Legend: had = Haden mango peel liqueur by alcoholic maceration, HDP = Haden mango peel liqueur by maceration with pectinase, TAA = Tommy Atkins mango peel liqueur by alcoholic maceration, TAP = Tommy Atkins mango peel liqueur by maceration with pectinase.

**Table 1 antioxidants-08-00102-t001:** Physicochemical analysis and color by *CIE L*a*b** system of mango peel liqueurs produced by different methods of maceration.

Liqueurs	Tommy Atkins		Haden	
Maceration Treatments	Alcoholic	Pectinase	Alcoholic	Pectinase
pH	4.85 ± 0.08 ^b^	3.61 ± 0.02 ^d^	5.01 ± 0.06 ^a^	3.94 ± 0.05 ^c^
Titratable acidity (g L^−1^)	1.0 ± 0.0 ^b^	4.4 ± 0.0 ^a^	1.1 ± 0.0 ^b^	4.6 ± 0.0 ^a^
Total sugars %	14.9 ± 0.1 ^a^	15.0 ± 0.1 ^a^	15.0 ± 0.1 ^a^	15.0 ± 0.2 ^a^
Alcoholic strength % (*v*/*v*)	18.2 ± 0.3 ^a^	18.0 ± 0.2 ^a^	17.9 ± 0.3 ^a^	18.0 ± 0.2 ^a^
Colour				
L*	59 ± 1 ^c^	62 ± 1 ^b^	55 ± 1 ^d^	70 ± 2 ^a^
a*	8.2 ± 0.2 ^a^	6.1 ± 0.2 ^b^	8.6 ± 0.1 ^a^	1.2 ± 0.1 ^c^
b*	80.7 ± 0.9 ^a^	71.5 ± 0.7 ^b^	63.3 ± 0.2 ^c^	37.3 ± 0.8 ^d^

^a–d^ Means followed by the same letters in the same lines do not differ by Tukey test at 5% probability of error.

**Table 2 antioxidants-08-00102-t002:** Bioactive compounds profile (µg 100 mL^−1^) of mango peel liqueurs produced by different methods of maceration.

Liqueurs	Tommy Atkins		Haden	
Maceration Treatments	Pectinase	Alcoholic	Pectinase	Alcoholic
FLAVANOLS				
(+)-Catechin	12.0 ± 0.1 ^c^	141.3 ± 2.3 ^a^	16.0 ± 0.0 ^b^	8.0 ± 0.1 ^d^
(−)-Epicatechin	22.0 ± 0.3 ^a^	14.0 ± 0.2 ^c^	13.3 ± 1.1 ^c^	20.0 ± 0.3 ^b^
(−)-Epicatechin gallate	406.7 ± 0.0 ^b^	776.7 ± 12.2 ^a^	266.2 ± 17.4 ^c^	718.7 ± 15.1 ^a^
(−)-Epigallocatechin gallate	64 ± 0 ^c^	141.7 ±17.5 ^a,b^	174 ± 14 ^a^	134 ± 2 ^b^
Procyanidin A_2_	36.7 ± 1.1 ^b^	78 ± 2 ^a^	22.0 ± 0.0 ^c^	14.0 ± 0.8 ^d^
Procyanidin B_1_	40 ± 0 ^c^	88 ± 1 ^a^	29.3 ± 1.1 ^d^	80 ± 1 ^b^
Procyanidin B_2_	ND	8.7 ± 1.1 ^a^	4.0 ± 0.1 ^b^	10.0 ± 0.1 ^a^
Total Flavanols quantification	581 ± 9	1254 ± 26	541 ± 30	950 ± 67
FLAVONOLS				
Kaempferol 3-glucoside	23.3 ± 1.1 ^c^	30.7 ± 2.3 ^b,c^	41.3 ± 9.2 ^a,b^	46.0 ± 0.1 ^a^
Myricetin	6.7 ± 1.1 ^c^	13.3 ± 1.1 ^b^	17.3 ± 2.3 ^a^	8.0 ± 0.1 ^c^
Isorhamnetin	11.3 ± 1.1 ^b^	24.0 ± 4.0 ^a^	20.7 ± 6.4 ^a,b^	30.7 ± 1.1 ^a^
Rutin	42.0 ± 0.2 ^b^	39.3 ± 1.1 ^b^	14 ± 2 ^c^	58.0 ± 0.1 ^a^
Quercetin 3-glucoside	14.0 ± 0.0 ^b^	6.0 ± 0.1 ^c^	22.0 ± 0.1 ^a^	6.0 ± 0.1 ^c^
Quercetin-3-*O*-glucopyranoside	343.3 ± 1.1 ^a^	14.0 ± 3.5 ^c^	348.7 ± 18.6 ^a^	42.7 ± 1.1 ^b^
Total Flavonols quantification	4401 ± 2	127 ± 5	464 ± 14	191 ± 1
STILBENE				
*trans*-Resveratrol	4.0 ± 0.3 ^a^	ND	2.0 ± 0.2 ^b^	2.0 ± 0.1 ^b^
PHENOLIC ACIDS				
Gallic acid	18.0 ± 8.3 ^d^	2271 ± 12 ^b^	1225.3 ± 12.8 ^c^	7512 ± 28 ^a^
Cinnamic acid	8.0 ± 0.1 ^c^	45.3 ± 9.4 ^b^	79.3 ± 2.3 ^a^	4.7 ± 1.1 ^c^
*p*-Coumaric acid	2.0 ± 0.1 ^b^	72 ± 4 ^a^	4.0 ± 0.3 ^b^	4.0 ± 0.1 ^b^
*o*-Coumaric acid	20.0 ± 0.2 ^c^	114 ± 6 ^b^	306.0 ± 14.4 ^a^	21.3 ± 1.1 ^c^
Benzoic acid	ND	ND	ND	1777.3 ± 7.6 ^a^
Syringic acid	96.0 ± 3.5 ^b^	391.3 ± 21.4 ^a^	ND	90 ± 2 ^b^
Total phenolics acids quantification	145 ± 9	2530 ± 45	1615 ± 29	9269 ± 2
Total phenolics quantification by HPLC	1167 ± 6	4303 ± 31	2622 ± 25	10523 ± 129
Total Phenolics ^§^	38,758 ± 133 ^b^	64,787 ± 170 ^a^	63,479 ± 116 ^a^	70,564 ± 186 ^a^

^a–d^ Means followed by the same letters in the same lines do not differ by Tukey test at 5% probability of error. ^§^ Total phenolics measured with Folin–Ciocateau method expressed as equivalent to gallic acid mg L^−1^.

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
