# Peer review of "Bioactive Compounds and Antioxidant Activity of Mango Peel Liqueurs (Mangifera indica L.) Produced by Different Methods of Maceration"

_antioxidants, 2019, doi:10.3390/antiox8040102_

Round 1
Reviewer 1 Report
The manuscript deals with the important topic of valorization of by-products. In particular, the peels of Mango represent an interesting and rich source of bioactives.
The experimental work is well conducted.
There are some concern regarding the presentation of the data and some the explanation given by the authors.
The manuscript is not always clear and needs to be revised for English language and mistyping.
Therefore, I recommend its acceptance after major revisions listed below.
Major:
1) Pag 3, line 135: authors should take care of decimal figures; “0.900 ± 0.050 and 0.700 ± 0.030” should be replaced by “0.90 ± 0.05 and 0.70 ± 0.03”
2) Pag 3, line 137: the measurement at t = 0 min is useless, unless more time points are considered to study kinetics
3) Pag 4, lines 152-154: authors quote “The lower values of pH and higher values of titratable acidity in the liqueurs obtained by maceration with pectinase can be explained by the higher solubility of organic acids of the peels in water”. Authors should clarify the nature of organic acids they are referring to, since the concentration of phenolic acids is by far higher in the samples extracted with ethanol than in those obtained with pectinase (as reported in Table 2). An accurate determination of specific organic acids (malic, tartaric, ascorbic, citric) would be therefore important. Please consider the following paper: Ma et al., Metabolites 2018, 8(4), 74.
4) Pag 4, Table 1: authors should take care of decimal figures; “14.9 ± 0.05”, “58.7± 1.1”, “15 ± 0.1”, “18 ± 0.2”, “62.3 ± 1.1”, “55.0 ± 1.3”, “70.0±1.7” should be replaced by “14.90 ± 0.05”, “59 ± 1”, “15.0 ± 0.1”, “18.0 ± 0.2”, “62 ± 1”, “55 ± 1”, “70 ± 2”, respectively.
5) Pag 5, Table 2: authors should take care of decimal figures here too.
Minor:
Pag 1, line 26: get rid of "it"
Pag 1, line 30: "the mango" should be replaced by "them"
Pag 1, line 33: author should modify the sentence “such diseases such as cancer, degenerative diseases, cardiovascular and diabetes” by "degenative diseases including cancer, cardiovascular diseases and diabetes"
Pag 2, lines 49-50: “a technique that can be used to obtain liqueurs because these beverages are obtained by the maceration of fruit pulp/peels” this sentence can be removed, the explanation of the use for the use of pectinase is better expressed in the following sentence.
Pag 6, line 200: “were considered acceptable, compared with” should be replaced by "were in line with"
Pag 6, line 210: “in” should be replaced by “with"
Pag 6, line 214: “gluciside” should be replaced by “glucoside"
Pag 6, line 221: “arabnoside” should be replaced by “arabinoside"
Pag 6, line 223: “in the values” should be replaced by “on the values "
Pag 7, lines 262-274: authors should re-write the sentence “indicating that berry liqueurs have high antioxidant potential, making them useful for alternative processes for the reuse of this by-product.”, since it is confusing and it is not clear that “this by-product” means mango peels.
Author Response
The manuscript deals with the important topic of valorization of by-products. In particular, the peels of Mango represent an interesting and rich source of bioactives.
The experimental work is well conducted.
There are some concern regarding the presentation of the data and some the explanation given by the authors.
The manuscript is not always clear and needs to be revised for English language and mistyping.
A general review of English was performed throughout the manuscript. (see lines 329-330)
Therefore, I recommend its acceptance after major revisions listed below.
Major:
1) Pag 3, line 135: authors should take care of decimal figures; “0.900 ± 0.050 and 0.700 ± 0.030” should be replaced by “0.90 ± 0.05 and 0.70 ± 0.03”
It was corrected (line 137) as asked by the Reviewer
2) Pag 3, line 137: the measurement at t = 0 min is useless, unless more time points are considered to study kinetics
It was corrected (line 139) as asked by the Reviewer
3) Pag 4, lines 152-154: authors quote “The lower values of pH and higher values of titratable acidity in the liqueurs obtained by maceration with pectinase can be explained by the higher solubility of organic acids of the peels in water”. Authors should clarify the nature of organic acids they are referring to, since the concentration of phenolic acids is by far higher in the samples extracted with ethanol than in those obtained with pectinase (as reported in Table 2). An accurate determination of specific organic acids (malic, tartaric, ascorbic, citric) would be therefore important. Please consider the following paper: Ma et al., Metabolites 2018, 8(4), 74.
It was corrected, as well observed by the Reviewer (see lines 156-157, 380).
4) Pag 4, Table 1: authors should take care of decimal figures; “14.9 ± 0.05”, “58.7± 1.1”, “15 ± 0.1”, “18 ± 0.2”, “62.3 ± 1.1”, “55.0 ± 1.3”, “70.0±1.7” should be replaced by “14.90 ± 0.05”, “59 ± 1”, “15.0 ± 0.1”, “18.0 ± 0.2”, “62 ± 1”, “55 ± 1”, “70 ± 2”, respectively.
It was corrected, as well observed by the Reviewer (see Table 1).
5) Pag 5, Table 2: authors should take care of decimal figures here too.
It was corrected, as well observed by the Reviewer (see Table 2).
Minor:
Pag 1, line 26: get rid of "it"
It was corrected (line 27) as asked by the Reviewer.
Pag 1, line 30: "the mango" should be replaced by "them"
It was corrected (line 31) as asked by the Reviewer.
Pag 1, line 33: author should modify the sentence “such diseases such as cancer, degenerative diseases, cardiovascular and diabetes” by "degenative diseases including cancer, cardiovascular diseases and diabetes"
Reviewer is right. This issue was corrected in line 34
Pag 2, lines 49-50: “a technique that can be used to obtain liqueurs because these beverages are obtained by the maceration of fruit pulp/peels” this sentence can be removed, the explanation of the use for the use of pectinase is better expressed in the following sentence.
Reviewer is right. This issue was corrected in line 49
Pag 6, line 200: “were considered acceptable, compared with” should be replaced by "were in line with"
This was corrected (line 203)
Pag 6, line 210: “in” should be replaced by “with"
This was corrected (line 214)
Pag 6, line 214: “gluciside” should be replaced by “glucoside"
This was corrected (line 217)
Pag 6, line 221: “arabnoside” should be replaced by “arabinoside"
This was corrected (line 224)
Pag 6, line 223: “in the values” should be replaced by “on the values "
This was corrected (line 226)
Pag 7, lines 262-274: authors should re-write the sentence “indicating that berry liqueurs have high antioxidant potential, making them useful for alternative processes for the reuse of this by-product.”, since it is confusing and it is not clear that “this by-product” means mango peels.
Reviewer is right. This issue was corrected in lines 267-268
Finally, we would like to thank the Reviewer comments and we believe that the final version of our manuscript may be of great interest to the readers of Antioxidants.
Reviewer 2 Report
This paper deals with bioactive compounds and antioxidant activity of mango peel liqueurs (Mangifera indica L.) produced by different methods of maceration. This manuscript has interest due to the importance of recovering an important part of the bioactive content of mango peels suggesting a suitable alternative process for the reuse of this by-product. I have carefully read the manuscript and there are some suggestions that the authors should consider to improve the way in which the manuscript is presented.
Stat of art, of the developed topic in this work, as well as the contribution to knowledge and purposes of this work should be included in Introduction. The author could also be more focused in the bioactive compounds and antioxidant activity of other liquors.
The author showed that with the use of mango peels to produce liqueurs it is possible to recover much of the bioactive content of the mango peels, which are one of the main by-products of the fruit industry. However, it will be important to evaluate also the sensory differences related to the different chemical composition observed by using different maceration processes and also different mango varieties. For example, if we have more catechin, probably the liquor will be more bitter or more astringent if we have more procyanidins. Therefore, it will be also important to evaluate the sensory profile and the consumer acceptance of the different liquors obtained with the different maceration techniques and mango varieties.
Author Response
This paper deals with bioactive compounds and antioxidant activity of mango peel liqueurs (Mangifera indica L.) produced by different methods of maceration. This manuscript has interest due to the importance of recovering an important part of the bioactive content of mango peels suggesting a suitable alternative process for the reuse of this by-product. I have carefully read the manuscript and there are some suggestions that the authors should consider to improve the way in which the manuscript is presented.
Stat of art, of the developed topic in this work, as well as the contribution to knowledge and purposes of this work should be included in Introduction. The author could also be more focused in the bioactive compounds and antioxidant activity of other liquors.
We added the reviewer's suggestion in the introduction, see lines 53-59
The author showed that with the use of mango peels to produce liqueurs it is possible to recover much of the bioactive content of the mango peels, which are one of the main by-products of the fruit industry. However, it will be important to evaluate also the sensory differences related to the different chemical composition observed by using different maceration processes and also different mango varieties. For example, if we have more catechin, probably the liquor will be more bitter or more astringent if we have more procyanidins. Therefore, it will be also important to evaluate the sensory profile and the consumer acceptance of the different liquors obtained with the different maceration techniques and mango varieties.
We appreciate the reviewer's suggestion, and we understand that comments are important to complement this work. However, in this first study we focused on measuring the differences obtained in the profile of phenolic compounds with antioxidant properties, in order to evaluate if the liquors presented potential as a functional beverage.
Finally, we would like to thank the Reviewer comments and we believe that the final version of our manuscript may be of great interest to the readers of Antioxidants.
Reviewer 3 Report
The present manuscript is an attractive and well-structured study, however it must be checked for English grammar. Furthermore, I have some comments in order to be suitable for publication.
Lines 70-71. Correct the “quercetin pyranoside” with “quercetin-3-O-glucopyranoside”.
Line 103. The term “classic analysis” is not appropriate, change as “physicochemical analysis”
Line 110. The FD have to be removed from the RP-HPLC/DAD/FD.
Line 147. Correct the title as “quality parameters”.
Lines 241-242. Add the units of ABTS and DPPH
Conclusions. You must emphasize that alcoholic maceration produced liqueurs with higher content of phytochemicals and antioxidant activity than maceration with pectinase.
Author Response
The present manuscript is an attractive and well-structured study, however it must be checked for English grammar. Furthermore, I have some comments in order to be suitable for publication.
Lines 70-71. Correct the “quercetin pyranoside” with “quercetin-3-O-glucopyranoside”.
This was corrected (lines 16, 74, and throughout the manuscript), as well observed by the Reviewer.
Line 103. The term “classic analysis” is not appropriate, change as “physicochemical analysis”
It was corrected, as asked by the Reviewer. (see line 106).
Line 110. The FD have to be removed from the RP-HPLC/DAD/FD.
The detection of phenolic compounds was performed on two detectors: diode array (DAD) and fluorescence (FD). See lines 115-116.
Line 147. Correct the title as “quality parameters”.
It was corrected, as asked by the Reviewer. (see line 150).
Lines 241-242. Add the units of ABTS and DPPH
It was corrected, as asked by the Reviewer. (see line 245-248).
Conclusions. You must emphasize that alcoholic maceration produced liqueurs with higher content of phytochemicals and antioxidant activity than maceration with pectinase.
It was corrected, as asked by the Reviewer. (see line 313-314).
Finally, we would like to thank the Reviewer comments and we believe that the final version of our manuscript may be of great interest to the readers of Antioxidants.
Round 2
Reviewer 1 Report
Authors have point-by-point replied to all the concerns raised previously. Therefore, in my opinion, the revised version of the manuscript merits publication in Antioxidants.
Reviewer 2 Report
abstract
line 16, 19, 73, 210, 216, 227, 289, 313, 317 and Table 2 - "quercetin-3-O-glucopyranoside" replce for "quercetin-3-O-glucopyranoside"
line 70 "procyanidin B1" replace to "procyanidin B1"
line 70 "procyanidin B2" replace to "procyanidin B2"
line 71 "procyanidin A2" replace to "procyanidin A2,"
line 79 "29.9 ºC" replace to "29.9 ºC"
Table 1 "colour" replace to "chromatic characteristics"
line 190, 213, 216 and 217 "100mL-1" replace to "100 mL-1"
line 202 "100g-1, replace to "100 g-1"
line 213 and 286 "trans-resveratrol" replace to "trans-resveratrol"